

# Tailored lymph node dissection in right hemicolectomy: a retrospective study focusing on the anterior tissue of the superior mesenteric vein surgical trunk

Xianda Chi[1], Xuejie Li[1], Qiong Liang[2], Pinjie Huang[3] and Jianpei Liu[1]

[1] Department of Gastrointestinal Surgery, The Third Affiliated Hospital of Sun Yat-sen University, Guangzhou, Guangdong, China
[2] Department of Pathology, The Third Affiliated Hospital of Sun Yat-sen University, Guangzhou, Guangdong, China
[3] Department of Anaesthesia, The Third Affiliated Hospital of Sun Yat-sen University, Guangzhou, Guangdong, China

Corresponding authors
Pinjie Huang, hpjie@126.com
Jianpei Liu, liujpei2@sysu.edu.cn

## ABSTRACT

**Background**. The optimal extent of lymph node dissection in right hemicolectomy for colon cancer remains a topic of debate. This study aimed to refine lymph node dissection strategies by investigating the histopathological characteristics of the anterior tissue of the superior mesenteric vein (SMV) surgical trunk.

**Methods**. One hundred sixty-two patients underwent surgery, with their medial resection border determined to be either to the right or left of the SMV. Pathological and perioperative variables were assessed, and the anterior tissue of the SMV was analyzed to quantify lymph nodes and nerve fibers.

**Results**. Of the patients included, 84 were in the SMV-right group and 78 in the SMV-left group. After propensity score matching (PSM), the SMV-left group with dissection extending to the left side of the SMV and removal of the anterior tissue of SMV surgical trunk, retrieved more lymph nodes (36.9 *vs.* 26.8, $P < 0.001$) than the SMV-right group. However, there was no difference in node-positive staging. The SMV-left group also experienced more postoperative complications (16.7% *vs.* 1.7%, $P = 0.011$) and prolonged postoperative defecation times (4.2 *vs.* 3.5, $P = 0.035$), accompanied by a higher resection of nerve fibers ($12.1 \pm 4.2$/case). Multivariate analysis identified tumor location above the ileocolic vein (ICV) root and elevated preoperative CA 19-9 levels as independent risk factors for metastasis to main lymph nodes.

**Conclusion**. Right hemicolectomy with extended lymph node dissection improves lymph node retrieval but increases complication risks and prolongs bowel recovery time. For patients with tumors located below the ICV root, a more limited dissection with the right side of the SMV as the medial boundary may be a preferable option, given the low rate of main lymph node metastasis.

## INTRODUCTION

Colon cancer (CC) is one of the most prevalent malignancies, ranking third in both incidence and mortality among male and female patients (*Siegel et al., 2023*). A comprehensive treatment regimen integrates surgery with adjuvant chemotherapy, immunotherapy, and targeted therapy, providing a spectrum of therapeutic options. In cases of advanced right-sided colon cancer, surgical intervention conventionally involves complete tumor excision coupled with routine lymph node dissection. However, there remains controversy regarding the medial border of lymph node dissection during right hemicolectomy surgery.

The current mainstream approaches to lymph node dissection are complete mesocolic excision (CME) and D3 lymph node dissection. A systematic review analyzing nine meta-analyses and three randomized trials indicates that CME surgery increases lymph node yield, improves survival rates, and decreases the risk of local and distant recurrence (*De Lange et al., 2023*). Similarly, *Kotake et al. (2014)* found that compared to D2 lymph node dissection, D3 lymph node dissection significantly improves overall survival in patients. CME involves high-level ligation of blood vessels (central vascular ligation, CVL), while Japanese D3 lymph node dissection is defined as dissection of the main lymph node "along the superior mesenteric vein and lateral to the superior mesenteric artery" (*Hashiguchi et al., 2020*). Both CME and D3 lymph node dissection surgeries advocate for placing the medial border of lymph node dissection on the left side of the superior mesenteric vein (SMV) (*Hashiguchi et al., 2020*; *Hohenberger et al., 2009*). However, proponents of traditional right hemicolectomy surgery argue that D2 lymph node dissection surgery is sufficient, and the medial border of lymph node dissection should be placed on the right side of the SMV. They contend that the central lymph node metastasis rate is only 2.1–3%, with a low incidence and no isolated central lymph node metastasis (CLM) (*Palmeri et al., 2022*). For T1 and T2 stage right-sided colon cancers, as well as T3 stage right-sided colon cancer with the primary tumor located more than seven cm from the main blood supply vessel, the rate of central lymph node metastasis is even lower (*Hida et al., 2005*). In addition, it is important to note that CVL or D3 lymph node dissection surgery can lead to denervation of the remaining intestinal tract, reduced intestinal motility, increased bowel frequency, affecting bowel function, and an increased incidence of lymphatic leakage and vascular injury (*Freund et al., 2016*; *Thorsen et al., 2021*).

The main difference between SMV-right and SMV-left lymph node dissection in right hemicolectomy lies in the exposure of the SMV surgical trunk and the resection of its anterior tissue (*Jin et al., 2006*). To our knowledge, there is no pathological evidence regarding the separate detection of the anterior tissue of the surgical trunk of the superior mesenteric vein. This study aims to identify and separately collect the anterior adipose lymphoid tissue of the SMV surgical trunk from the excised specimens, submit it for individual pathological testing, and gather data on the total number of lymph nodes, the number of metastatic lymph nodes, and the number of nerve fibers present in this tissue. The clinical significance of exposing the surgical trunk of SMV and resecting its anterior tissue during right hemicolectomy will be evaluated from a clinicopathological

perspective. Moreover, the study will assess the correlation between the existing nerve fibers in the anterior tissue of the SMV surgical trunk and the recovery of postoperative bowel function. The ultimate goal of this research is to provide evidence and justification for the individualized determination of the extent of lymph node dissection during right hemicolectomy.

## METHODS

### Patient selection and data collection

Patients who underwent colectomy and lymphadenectomy for Stage I–III right-sided colon cancer at the Third Affiliated Hospital of Sun Yat-sen University between January 2017 and December 2018, and between January 2022 and December 2024, were reviewed. This study has been approved by the ethics committee of Third Affiliated Hospital of Sun Yat-sen University, code number: SL-II2024-213-01. The inclusion criteria were as follows: (1) the endoscopic biopsy diagnosis of the primary lesion in the colon was adenocarcinoma; (2) the preoperative clinical stages were T1-4, N0-2, M0 (AJCC-8th); and (3) the surgery was performed, achieving an R0 resection result. The exclusion criteria were as follows: (1) the patient underwent emergency surgery; (2) the tumor had extensively invaded surrounding tissues, making an R0 resection unattainable; (3) the patient required simultaneous surgery for concurrent diseases; and (4) preoperative investigations did not reveal evidence of distant metastasis, while intraoperative exploration or postoperative pathology confirmed distant metastasis.

Patients included between January 2017 and December 2018 underwent traditional right hemicolectomy, with lymphadenectomy extending to the right border of the superior mesenteric vein (SMV). These patients were defined as the SMV-right group. After 2019, central vascular ligation (CVL) became the broadly accepted gold standard for right hemicolectomy in our department. Therefore, patients included between January 2022 and May 2024 underwent right hemicolectomy with lymphadenectomy extending to the left border of the SMV and were defined as the SMV-left group.

The information obtained from medical records included age, gender, body mass index (BMI), American Society of Anesthesiologists (ASA) grade, comorbidities, tumor location, pathologic TNM stage, operation approach, preoperative carcinoembryonic antigen (CEA) and CA19-9 levels, operation duration, blood loss, surgical approach, surgical method, surgical time, blood loss, defecation time, days to drain removal, postoperative drainage volume, and postoperative complications, which included poor wound healing, chylous fistula, abdominal infection, abdominal bleeding, and ileus. These data were recorded and analyzed. It is worth mentioning that, based on the Alinity I system (Abbott Laboratories, Abbott Park, IL, USA) used at our center, we adopted a dichotomous cutoff of 35 ng/ml for CA 19-9 values. Additionally, the term 'days to drain removal' refers to the days it takes for an abdominal cavity drainage tube to be removed after surgery.

### Surgical procedure

Right hemicolectomy was performed carefully separating the embryological visceral and parietal planes. The lateral-to-medial, medial-to-lateral, and posterior approaches to

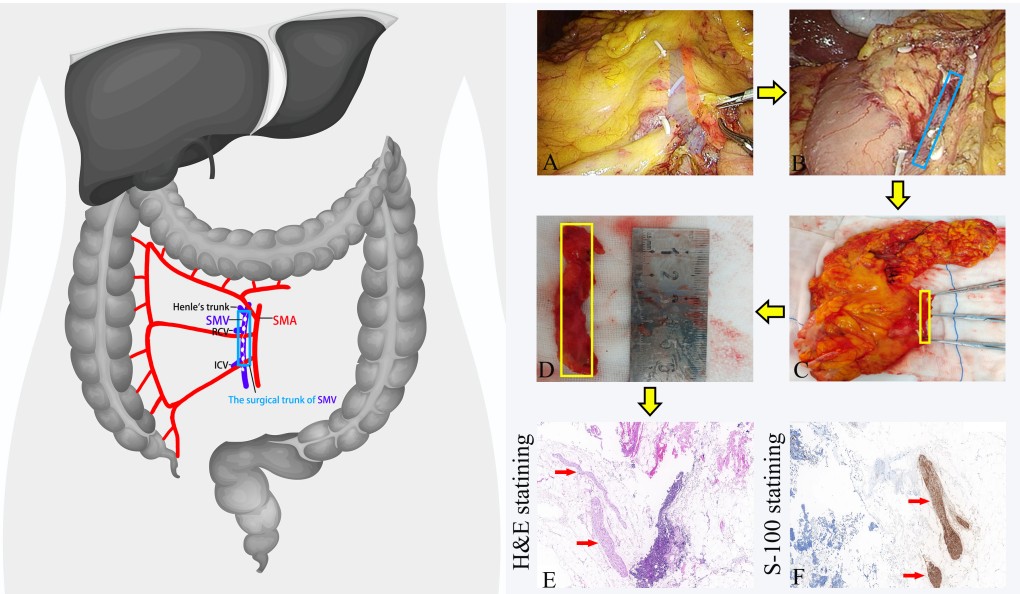

**Figure 1** The procedures for dissecting the anterior tissue of the surgical trunk of the superior mesenteric vein (SMV) during right hemicolectomy. (A) Dissection of the anterior tissue of the SMV surgical trunk during the operation; (B) complete exposure of the SMV during the operation; (C) dissection of the anterior tissue of the SMV surgical trunk *in vitro*; (D) the separated anterior tissue of the SMV surgical trunk being measured with a ruler. (E) The anterior tissue of the SMV surgical trunk after H&E staining. The nerve fibers are highlighted by red arrows. (F) The anterior tissue of the SMV surgical trunk after S-100 staining. The nerve fibers are displayed in brown and highlighted by red arrows. SMA, Superior mesenteric artery. SMV, Superior mesenteric vein. RCV, right colon vein. ICV, ileocolic vein.

dissection were utilized, developing a plane between the white line of Toldt and Gerota's fascia. Vessels were ligated close to their origins. The surgical procedures were categorized as follows: resection of the ileocolic artery (ICA), the right colic artery (RCA), and the right branch of the middle colic artery (MCA) were defined as right hemicolectomy; and resection of the left branch of the MCA in addition to these was defined as extended right hemicolectomy. The minimum bowel resection margin, measuring 15 cm from the tumor, was set on both the oral and anal sides. For the SMV-left group, the tissue anterior to the superior mesenteric vein (SMV) was thoroughly dissected to achieve complete exposure of the SMV. In contrast, for the SMV-right group, the area in front of the SMV was not intentionally cleaned, resulting in incomplete exposure of the SMV (Fig. 1).

## Pathological specimen processing

After surgery, a colorectal surgeon classified lymph nodes based on their positions within the resected specimens. The extent of lymph node (LN) dissection encompassed pericolic LNs along the marginal arteries, intermediate LNs along the main feeding artery, and main nodes along the superior mesenteric vessels. The specimen was processed using standard pathological methods and subsequently reviewed by a single, specialized pathologist.

In the SMV-left group, we submit it for individual examination after isolating the anterior tissue of the surgical trunk of the superior mesenteric vein (SMV) *in vitro*. The

surgical trunk is defined as the SMV from the confluence of the ileocolic vein (ICV) to Henle's trunk (*Jin et al., 2006*). We defined the anterior tissue of the SMV surgical trunk as follows: the upper boundary was at the level of the root of Henle's trunk, the lower boundary was at the level of the root of the ileocolic vein (ICV), with the lateral boundaries being the left and right sides of the SMV, respectively. Subsequently, the tissue was fixed in formalin solution, dehydrated in ethanol, used the whole mount technique, and then cut into thin slices with a thickness of 4 um. After the slices were prepared into pathology slides, they underwent hematoxylin and eosin staining (H&E-staining). Experienced pathologists recorded the number of lymph nodes and positive lymph nodes in each specimen under a microscope. Additionally, in each case, we selected representative slides for S-100 staining to record the number of nerve fibers under the microscope (Fig. 1).

### Data statistics

Quantitative data were expressed as mean $\pm$ SD. Qualitative data were expressed as the frequency and percentage. The $t$-test was used to compare continuous variables. The chi-square test was used for categorical variables. Logistic regression analysis was performed to estimate the risk of lymph node metastasis, which was adjusted for sex, age, BMI, tumor location, operation method, preoperative CEA, and CA19-9 levels. A $P$ value of <0.05 indicated a statistically significant difference. Statistical analysis was performed using SPSS (version 27.0; IBM Corporation).

To enhance the homogeneity of baseline data, we employed propensity score matching (PSM) to adjust for differences in baseline characteristics between the 2 groups. This adjustment was performed using SPSS (version 27.0; IBM Corporation) with a combination of one-to-one pairing and the nearest neighbor methods with a caliper of 0.02. The PSM process involved the incorporation of key variables, including age, sex, BMI score, tumor location, pathologic TNM stage and operation approach.

## RESULTS

### Patient characteristics

Of the patients included, 84 were in the SMV-right group and 78 in the SMV-left group. The patients were divided into two groups based on the different approaches to lymphadenectomy: the SMV-right group ($n = 84$) and the SMV-left group ($n = 78$). The selection process is described in the Methods section. After PSM analysis, a total of 120 patients (60 patients from the SMV-right group and 60 patients from the SMV-left group) were included in this study. No significant differences were found in the clinicopathological characteristics between the two groups (Table 1).

### Short-term outcomes

Compared to the SMV-right group, patients in the SMV-left group experienced longer postoperative defecation times, more days to drain removal, and higher postoperative drainage volume. Additionally, they had a higher overall incidence of postoperative complications, including poor wound healing, chylous fistula, and abdominal infection. However, when analyzing the rate of each of these complications individually, no statistically

**Table 1  Clinical characteristics of patients in the two groups.**

| Variable | Before PSM | | | After PSM | | |
|---|---|---|---|---|---|---|
| | SMV-right group (*n* = 84) | SMV-left group (*n* = 78) | *P*-value | SMV-right group (*n* = 60) | SMV-left group (*n* = 60) | *P*-value |
| Sex | | | 0.673 | | | 0.855 |
| Male | 48 | 42 | | 31 | 32 | |
| Female | 36 | 36 | | 29 | 28 | |
| Age (years) (mean ± SD) | 63.6 ± 13.7 | 62.8 ± 13.3 | 0.707 | 62.4 ± 14.1 | 62.5 ± 13.4 | 0.958 |
| BMI (kg/m2) (mean ± SD) | 22.3 ± 3.3 | 21.9 ± 3.3 | 0.419 | 22.0 ± 3.3 | 22.3 ± 3.3 | 0.587 |
| ASA grades | | | 0.271 | | | 0.378 |
| I | 13 | 9 | | 9 | 9 | |
| II | 48 | 52 | | 35 | 37 | |
| III | 20 | 17 | | 13 | 14 | |
| IV | 3 | 0 | | 3 | 0 | |
| Tumor location | | | 0.212 | | | 0.985 |
| Cecum | 15 | 9 | | 9 | 8 | |
| Ascending colon | 40 | 50 | | 36 | 36 | |
| Hepatic flexure | 19 | 12 | | 10 | 10 | |
| Transverse colon | 10 | 7 | | 5 | 6 | |
| Operation approach | | | 0.130 | | | NS |
| Open | 8 | 2 | | 2 | 2 | |
| Laparoscopy | 76 | 76 | | 58 | 58 | |
| Scope of surgery | | | 0.36 | | | 0.471 |
| Right hemicolectomy | 64 | 64 | | 51 | 48 | |
| Extended right hemicolectomy | 20 | 14 | | 9 | 12 | |
| Pathologic TNM_stage | | | 0.431 | | | 0.933 |
| I | 17 | 17 | | 12 | 11 | |
| II | 35 | 25 | | 21 | 20 | |
| III | 32 | 36 | | 27 | 29 | |
| Pathologic N_stage | | | 0.486 | | | 0.920 |
| N0 | 52 | 42 | | 33 | 31 | |
| N1 | 20 | 25 | | 18 | 20 | |
| N2 | 12 | 11 | | 9 | 9 | |
| Preoperative CEA levels (ng/ml) | | | 0.327 | | | 0.13 |
| ≤5 | 52 | 54 | | 34 | 42 | |
| >5 | 32 | 24 | | 26 | 18 | |
| Preoperative CA19-9 levels (ng/ml) | | | 0.988 | | | 0.5 |
| ≤35 | 69 | 64 | | 46 | 49 | |
| >35 | 15 | 14 | | 14 | 11 | |
| Preoperative bowel obstruction | | | 0.677 | | | 0.527 |
| Yes | 9 | 10 | | 4 | 7 | |
| No | 75 | 68 | | 56 | 53 | |

**Table 2 Short-term outcomes of patients in two groups.**

| Variable | Before PSM | | | After PSM | | |
|---|---|---|---|---|---|---|
| | SMV-right group (n = 84) | SMV-left group (n = 78) | P-value | SMV-right group (n = 60) | SMV-left group (n = 60) | P-value |
| Operation duration (min) (mean ± SD) | 220.1 ± 48.2 | 224.4 ± 54.0 | 0.593 | 213.6 ± 44.9 | 225.1 ± 55.9 | 0.218 |
| Blood loss (ml) (mean ± SD) | 67.0 ± 44.3 | 57.9 ± 51.2 | 0.231 | 60.1 ± 41.6 | 60.8 ± 52.0 | 0.938 |
| Postoperative defecation time (day) (mean ± SD) | 3.6 ± 1.4 | 4.1 ± 1.9 | 0.048 | 3.5 ± 1.6 | 4.2 ± 1.9 | 0.035 |
| Postoperative hospital stay length (day) (mean ± SD) | 9.9 ± 7.4 | 8.5 ± 4.8 | 0.151 | 9.0 ± 3.5 | 8.7 ± 5.3 | 0.744 |
| Days to drain removal (day) (mean ± SD) | 3.3 ± 2.1 | 5.3 ± 3.5 | <0.001 | 3.0 ± 1.7 | 5.3 ± 3.9 | <0.001 |
| Post-operative drainage volume(ml) (mean ± SD) | 489.1 ± 635.8 | 690.7 ± 550.6 | 0.033 | 359.0 ± 307.4 | 673.7 ± 558.2 | <0.001 |
| Postoperative diarrhea | | | 0.234 | | | 0.306 |
|   Yes | 14 | 8 | | 11 | 7 | |
|   No | 70 | 70 | | 49 | 53 | |
| Post-operative complications | | | | | | |
|   Poor wound healing | 2 (2.4%) | 3 (3.8%) | 0.933 | 1 (1.7%) | 3 (5.0%) | 0.611 |
|   Chylous fistula | 2 (2.4%) | 2 (2.6%) | NS | 0 | 2 (3.3%) | 0.476 |
|   Abdominal infection | 1 (1.2%) | 5 (6.4%) | 0.180 | 0 | 3 (5.0%) | 0.242 |
|   Abdominal bleeding | 0 | 1 (1.3%) | 0.970 | 0 | 1 (1.7%) | NS |
|   Ileus | 0 | 1 (1.3%) | 0.970 | 0 | 1 (1.7%) | NS |
|   Total | 5 (6.0%) | 12 (15.4%) | 0.05 | 1 (1.7%) | 10 (16.7%) | 0.011 |

significant differences were found between the two groups. It is worth mentioning that one patient in the SMV-left group suffered from SMV injury during surgery, while none of the patients suffered from SMA injury (Table 2).

## Pathological outcomes

All pathological specimens resected from patients were reviewed by specialized pathologists after H&E staining. The pathological staging for all patients was based on the AJCC 8th edition. No distant metastases were observed in any patient in either group. No distant metastases were observed in any patient in either group. There were no significant differences between the two groups in terms of pathologic T stage and pathologic N stage.

After PSM, the SMV-left group demonstrated a significantly higher total number of harvested lymph nodes (LNs) compared to the SMV-right group (36.9 vs. 26.8, $P < 0.001$). Similarly, the number of main LNs was also significantly greater in the SMV-left group (16.6 vs. 9.0, $P < 0.001$). A subtle difference was observed in the number of intermediate lymph nodes between the two groups after PSM ($P = 0.049$). In contrast, no significant differences were found in the number of paracolic LNs between the groups. Additionally, no significant difference was observed in either the total number of positive LNs or the number of positive LNs in the paracolic, intermediate, and central areas between the two

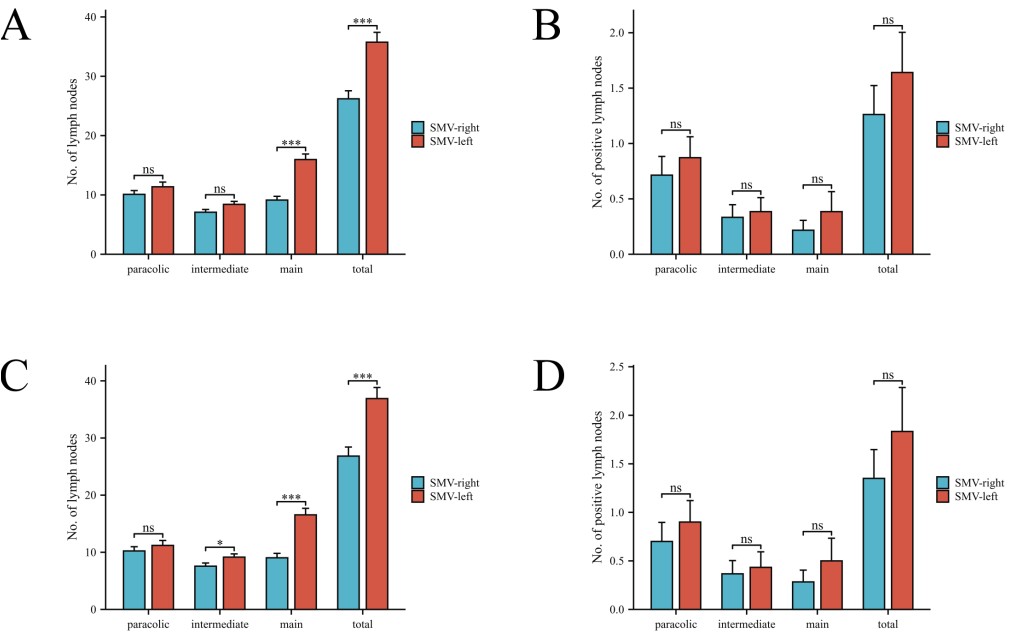

**Figure 2 Comparison of lymph node counts and positive lymph nodes in different regions between two patient groups.** (A, B) Comparison before PSM. (C, D) Comparison after PSM.

groups. Notably, in the SMV-left group, two cases of lymph node metastasis were found in the anterior tissue of the surgical trunk of the superior mesenteric vein (SMV). One patient's tumor location was in the ascending colon, another was in the hepatic flexure. Both cases' preoperative clinical TNM staging was T4N2M0, and no skip metastases occurred. The pathological outcomes are illustrated in Fig. 2.

The mean length the anterior tissue of the SMV surgical trunk was $4.8 \pm 1.5$ cm. After H&E and S-100 stained, we found $4.1 \pm 4.0$ lymph nodes and $12.1 \pm 4.2$ nerve fibers average in each tissue.

## Multivarible regression analysis

In order to identify risk factors for lymph node metastasis in the main lymph nodes, we further analyzed patients' CT images and regrouped them based on the relationship between the tumor location and the root level of the ileocolic vein (ICV) in the coronal plane. This grouping was based on the fact that the ICV root serves as the lower boundary of the surgical trunk of the superior mesenteric vein (SMV). A total of 75 patients were placed in the Below ICV group, while 87 patients were placed in the Above ICV group. In the Below ICV group, 25 (33.3%) patients exhibited lymph node metastasis in the paracolic lymph nodes, 6 (8.0%) patients had metastasis in the intermediate lymph nodes, and 3 (4.0%) patients had metastasis in the main lymph nodes. In contrast, in the Above ICV group, 25 (28.7%) patients had lymph node metastasis in the paracolic lymph nodes, 19 (21.8%) patients had metastasis in the intermediate lymph nodes, and 13 (14.9%) patients had metastasis in the main lymph nodes. Additionally, we found that 10 patients (6.2%)

had skip metastases to the main lymph node, with one patient in the Above ICV group and nine patients in the Below ICV group. By comparing the incidence of lymph node metastasis between the two groups, we observed that patients with primary lesions located above the ICV root were significantly more likely to have metastasis in the intermediate lymph nodes (21.8% *vs.* 8.0%, $P = 0.015$) and the main lymph nodes (14.9% *vs.* 4.0%, $P = 0.039$). Notably, we observed that patients with primary lesions located above the ICV root were significantly more likely to exhibit skip metastases to the main lymph nodes (10.3% *vs.* 1.3%, $P = 0.04$) (Fig. 3).

Afterwards, we incorporated this risk factor alongside age, sex, BMI, ASA grades, pathological T stage, preoperative CEA levels, and preoperative CA19-9 levels into a logistic regression analysis. Following both univariate and multivariate logistic regression, we discovered that tumors situated above the level of the ICV root (OR = 4.310, $P = 0.029$) and elevated preoperative CA 19-9 levels (OR = 3.309, $P = 0.039$) are independent risk factors for lymph node metastasis in the main lymph nodes. Furthermore, we discovered that tumors located above the level of the ICV root are an independent risk factor for skip metastases to the main lymph nodes (OR = 15.118, $P = 0.019$) (Fig. 4).

## DISCUSSION

The present study found that setting the left side of the superior mesenteric vein as the medial border for lymph node dissection in the right hemicolectomy and resecting the anterior tissue of the SMV surgical trunk increased lymph node harvest and positive LN harvest. Still, it did not increase the number of patients with metastatic lymph nodes or alter the pathological stage of any patients. In comparison to the SMV-right group, extending the medial border of lymph node dissection to the left side of the SMV in the SMV-left group increased postoperative complication and drain removal of the surgical site, and delayed the recovery of bowel function. Preoperative abnormal CA199 levels and tumor locations above the root level of the ileocolic vein (ICV) in the coronal plane were independent risk factors for main lymph node metastasis.

Currently, D3 lymph node dissection in Japan and CME surgery in Western countries are considered the mainstream surgical procedures for lymph node dissection in advanced right-sided colon cancer. Following the discovery by *Hohenberger et al. (2009)* that CME can significantly improve long-term survival rates for patients, *West et al. (2010)* further found that, compared to traditional procedures, CME+CVL can achieve a higher yield of lymph nodes. Compared with total mesorectal excision (TME), the promotion and application of CME are relatively slow in Western countries, which may be due to the conflicts in different research results and potential surgical risks (*Bertelsen et al., 2016*; *Freund et al., 2016*). Large-scale randomized controlled trials (RCTs) are still needed to provide evidence in support of the CME theory. Similarly, the D3 lymph node dissection faces a similar situation. On the other hand, *Bertelsen et al. (2016)* found that, compared to traditional surgical methods, CME+CVL is associated with higher rates of intraoperative losses and postoperative complications, particularly damage to the superior mesenteric vein (SMV) during surgery (1.7% *vs.* 0.2%, $p < 0.001$). Therefore, it is crucial to weigh the

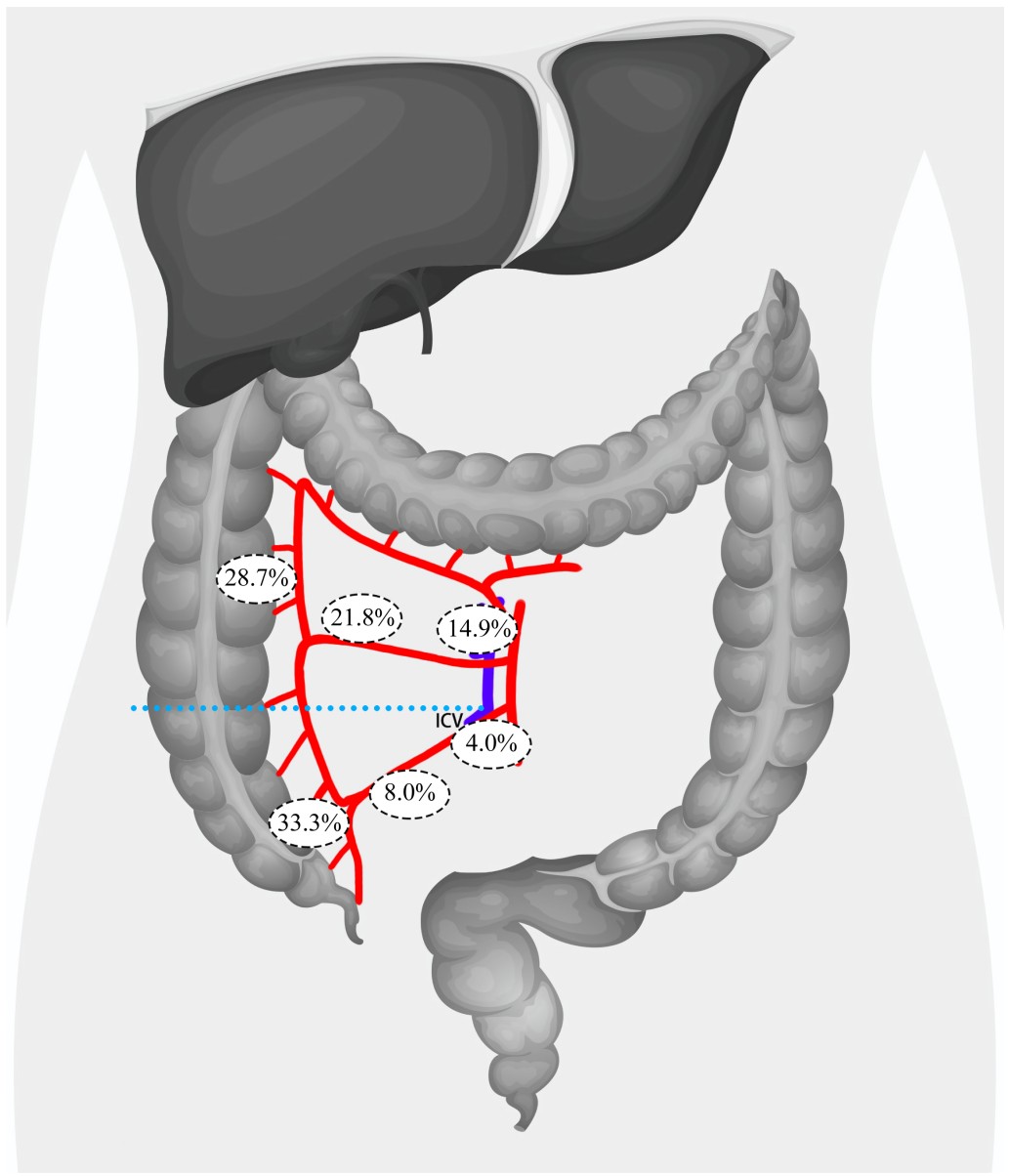

**Figure 3** **Distribution of lymph node metastasis rates across paracolic, intermediate, and main lymph node regions in patients with tumors located above or below the root level of ICV.** ICV, ileocolic vein.

pros and cons, as well as the long-term survival rates and short-term complications, and to individualize lymph node dissection methods.

The crucial difference between CME/D3 lymph node dissection and traditional right hemicolectomy lies in the exposure of the superior mesenteric vein (SMV) surgical trunk and the clearance of adipose lymphoid tissue in front of it. After separately detecting the anterior tissue of the SMV surgical trunk, our study found that, compared to traditional right hemicolectomy, using the left side of the SMV as the boundary for lymph node

Multivariable logistic regression analysis of predictors of main lymph node metastasis.

| Characteristics | Total(N) | OR(95% CI) | P value |
|---|---|---|---|
| Age | 162 | 1.012(0.972-1.053) | 0.562 |
| Sex | 162 | | |
| Female | 72 | Reference | |
| Male | 90 | 1.032(0.365-2.920) | 0.953 |
| BMI | 162 | 1.087(0.929-1.271) | 0.297 |
| ASA grades | 162 | | |
| I | 24 | Reference | |
| II | 96 | 0.404(0.093-1.760) | 0.228 |
| III&IV | 42 | 1.343(0.310-5.817) | 0.693 |
| pT_stages | 162 | | |
| T1&T2 | 39 | Reference | |
| T3&T4 | 123 | 5.278(0.674-41.315) | 0.113 |
| Tumor location | 162 | | |
| Below ICV | 75 | Reference | |
| Above ICV | 87 | 4.31(1.161-15.998) | 0.029 |
| Preoperative CEA levels(ng/ml) | 162 | | |
| ≤5 | 106 | Reference | |
| >5 | 56 | 1.768(0.605-5.164) | 0.297 |
| Preoperative CA19-9 levels(ng/ml) | 162 | | |
| ≤35 | 133 | Reference | |
| >35 | 29 | 3.309(1.060-10.325) | 0.039 |

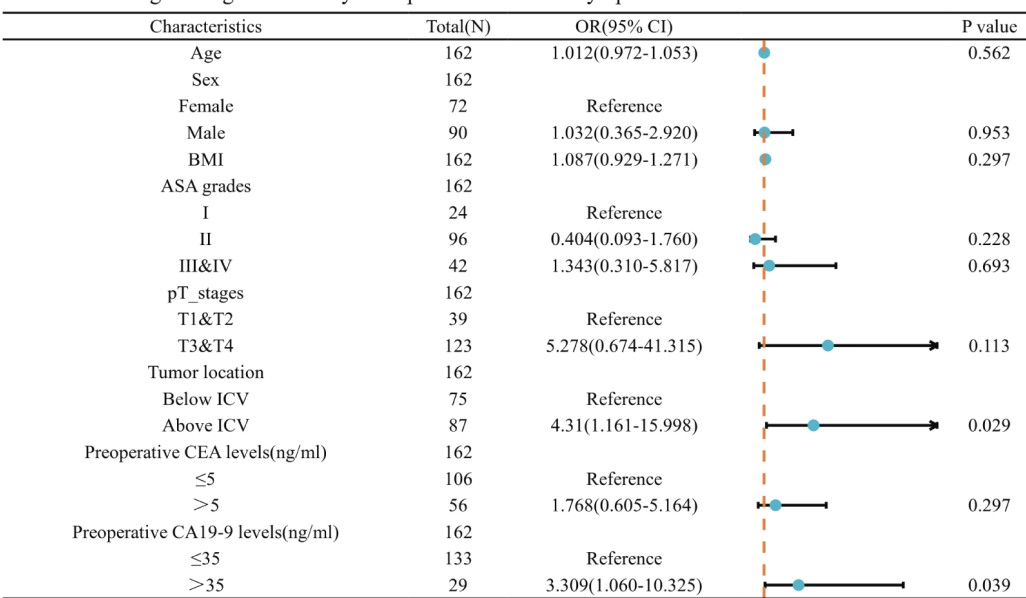

Multivariable logistic regression analysis of predictors of skip metastases to the main lymph node

| Characteristics | Total(N) | OR(95% CI) | P value |
|---|---|---|---|
| Age | 162 | 0.982(0.923-1.043) | 0.55 |
| Sex | 162 | | |
| Female | 72 | Reference | |
| Male | 90 | 0.731(0.171-3.125) | 0.673 |
| BMI | 162 | | |
| ASA grades | 162 | | |
| I | 24 | Reference | |
| II | 96 | 0.200(0.020-1.973) | 0.168 |
| III&IV | 42 | 1.063(0.085-13.316) | 0.962 |
| pT_stages | 162 | | |
| T1&T2 | 39 | Reference | |
| T3&T4 | 123 | 2.465(0.259-23.450) | 0.432 |
| Tumor location | 162 | | |
| Below ICV | 75 | Reference | |
| Above ICV | 87 | 15.118(1.565-146.004) | 0.019 |
| Preoperative CEA levels(ng/ml) | 162 | | |
| ≤5 | 106 | Reference | |
| >5 | 56 | 1.394(0.275-7.069) | 0.688 |
| Preoperative CA19-9 levels(ng/ml) | 162 | | |
| ≤35 | 133 | Reference | |
| >35 | 29 | 1.781(0.316-10.022) | 0.513 |

**Figure 4  Multivariable logistic regression analysis of predictors of main lymph node metastasis.** BMI, body mass index; ASA grades, American Society of Anesthesiologists grades; ICV, ileocolic vein.

dissection yields more lymph nodes, consistent with previous research findings (*West et al., 2010*). However, the number of patients with positive lymph nodes did not increase, and no isolated central LNM was found. Moreover, individual examination of the anterior tissue of the SMV surgical trunk revealed only two cases with positive lymph nodes after

H&E staining. This suggests that expanding the lymph node dissection range only increases lymph node yield without altering the N-stage of patients. Nevertheless, some studies have indicated that increasing lymph node yield can improve patient survival rates (*Chang et al., 2007*; *Chapuis et al., 1985*). Other researchers have discovered a correlation between the yield of negative lymph nodes and long-term survival rates in Stage III colon cancer. For example, *Kuo et al. (2022)* found that, in patients with right-sided colon cancer, those with a high yield of negative lymph nodes had significantly better 5-year overall survival (74.9% *vs.* 62.7%, $P < 0.001$) and 5-year recurrence-free survival (75.0% *vs.* 61.9%, $P < 0.001$) compared to patients with a low negative lymph node count (<27). Recent studies have revealed that, in addition to the anatomical pathway for lymph node metastasis (paracolic LNs → intermediate LNs → main LNs), there is a feature of "skip metastasis." In Stage III colon cancer patients, this proportion reaches 19.8% (*Liang et al., 2015*). Our study found that 19 patients (11.7%) exhibited skip metastasis, similar to the findings of the aforementioned research.

Postoperative complications following right-sided colon cancer lymph node dissection have been a noteworthy topic of discussion. A total of 17 patients (10.5%) in the present study experienced postoperative complications, with a significant difference in the incidence of complications between the two groups. Previous studies have also indicated that compared to traditional dissection methods, D3+CME may increase the incidence of postoperative complications (*Bertelsen et al., 2016*; *Ramser et al., 2021*), possibly due to the higher technical difficulty and demands on the surgical team. Moreover, this study found that, compared to the SMV-right group, the SMV-left group had longer drain removal and a higher postoperative drainage volume, which may be attributed to the larger surgical wound caused by expanding the lymph node dissection range, leading to increased wound exudation. Additionally, we also observed that patients who underwent denudation of the SMV had a longer postoperative defecation time. Upon S-100 staining of the anterior tissue of the SMV surgical trunk, we identified a significant amount of nerve fibers. The removal of excessive intrinsic nerve fibers supplying the small intestine in the SMV-left group, which leads to postoperative denervation, may partially contribute to the longer recovery time of bowel function in these patients.

Through multivariable logistic regression analysis, we have identified that tumors located above the root level of the ileocolic vein (ICV) in the coronal plane are an independent risk factor for central/main LN metastasis. We speculate that this may be due to the longer distance of tumors located in the cecum and lower part of the ascending colon from the superior mesenteric vein (SMV), resulting in a lower probability of tumor metastasis to the main LNs. According to the definition by the Japanese Society for Cancer of the Colon and Rectum (JSCCR), D3 lymphadenectomy involves the main LNs, including No. 203, No. 213, and No. 223, which are grouped according to their locations at the ileocolic artery (ICA), right colic artery (RCA), and middle colic artery (MCA), respectively (*Hashiguchi et al., 2020*). Previous studies have indicated a correlation between LN metastasis and tumor location in right-sided colon cancer, with tumors in the cecum region commonly metastasizing along the ICA and those in the ascending colon and hepatic flexure region along the RCA (*Park et al., 2009*). *Toyota, Ohta & Anazawa (1995)* found that malignant

tumors in the cecum region had slightly lower LN metastasis rates for No. 213 and No. 223 compared to other locations of right-sided colon cancer, with subsequent studies yielding similar results (*Lan et al., 2011*; *Tsukamoto et al., 2023*). An autopsy study revealed that the MCA region had more lymph nodes and shorter lymphatic vessels compared to the ICA region, which connects to lymph nodes in front of the SMV and superior mesenteric artery (SMA), indicating a higher rate of main LNs metastasis for tumors supplied by the MCA (*Nesgaard et al., 2018*). Furthermore, we found that tumors located above the root level of the ICV in the coronal plane are an independent risk factor for skip metastasis to the central/main lymph nodes (LNs). Therefore, we suggest that extended lymph node (LN) dissection may be the preferred approach for patients with tumors located above the root level of the ICV in the coronal plane, given the high rate of apical lymph node metastasis. However, for patients with tumors situated below the ICV root level in the coronal plane and exhibiting lymphatic drainage to the ICA region, a more limited LN dissection may be advantageous. This tailored approach could potentially enhance postoperative bowel function recovery and reduce the risk of complications, making it a clinically valuable recommendation.

This study has several limitations. Firstly, the timing of surgeries differed between the two groups. Although the same group of surgeons performed the surgeries, their surgical techniques may have changed over time. This could introduce bias into the study. In addition, the current research only compared short-term prognosis data for the two groups of patients undergoing different surgical methods, lacking a comparison of long-term prognosis data. Furthermore, a long-term follow-up is necessary to confirm our findings and provide a more comprehensive understanding of the outcomes.

## CONCLUSION

Right hemicolectomy with extended lymph node dissection offers benefits over traditional colectomy by improving the retrieval of lymph nodes and clearance of positive metastatic lymph nodes along the SMV surgical trunk. However, it also comes with higher risks of complications and slower bowel function recovery. Therefore, it may not be suitable for all patients with right-sided colon cancer. For patients with tumors located below the root level of the ICV in the coronal plane, a more limited lymph node dissection, with the right side of the superior mesenteric vein serving as the medial boundary, may be a preferable option. This approach is supported by the low rate of main lymph node metastasis in front of the SMV surgical trunk. It can enhance postoperative recovery and achieve the goals of individualized therapy.

### Funding

This work was supported by the Natural Science Foundation of Guangdong Province (No. 2023A1515010347 and 2021A1515011827). The funders had no role in study design, data collection and analysis, decision to publish, or preparation of the manuscript.

### Grant Disclosures

The following grant information was disclosed by the authors:
Natural Science Foundation of Guangdong Province: 2023A1515010347, 2021A1515011827.

### Competing Interests

The authors declare there are no competing interests.

### Author Contributions

- Xianda Chi performed the experiments, analyzed the data, prepared figures and/or tables, authored or reviewed drafts of the article, and approved the final draft.
- Xuejie Li performed the experiments, analyzed the data, prepared figures and/or tables, and approved the final draft.
- Qiong Liang performed the experiments, prepared figures and/or tables, and approved the final draft.
- Pinjie Huang conceived and designed the experiments, authored or reviewed drafts of the article, and approved the final draft.
- Jianpei Liu conceived and designed the experiments, authored or reviewed drafts of the article, and approved the final draft.

### Human Ethics

The following information was supplied relating to ethical approvals (i.e., approving body and any reference numbers):

This study has been approved by the ethics committee of Third Affiliated Hospital of Sun Yat-sen University (SL-II2024-213-01).

### Data Availability

The raw measurements are available in the Supplementary File.

### Supplemental Information

Supplemental information for this article can be found online at http://dx.doi.org/10.7717/peerj.19290#supplemental-information.

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
