# Peer review of "Tailored lymph node dissection in right hemicolectomy: a retrospective study focusing on the anterior tissue of the superior mesenteric vein surgical trunk"

_PeerJ, doi:10.7717/peerj.19290_

## Round 0.1 · original submission · Major Revisions

Dear authors,
thank your for your submission. Currently, significant revisions are advised. Namely:
- Ensure that Figure 1 is explicitly referred to in the main text and label sections E and F appropriately.
- Verify that all figures meet the minimum required resolution (DPI). The current figures may lack sufficient quality for publication. Please confirm and re-upload high-resolution versions as needed.
- At least for immunohistochemistry (IHC) figures, it is recommended to upload raw data for clarity. Additionally, highlight key observations in the figures to ensure readers can easily interpret critical findings.
- Include raw data on neural tissue in the area anterior to the superior mesenteric vein (SMV) to align with the study’s hypothesis.
- Provide a detailed explanation for using 35 ng/ml as the dichotomous cutoff for CA 19-9 values.
- Consider performing a propensity-matched analysis to adjust for age, gender, tumor stage, and primary tumor location. This would strengthen the validity of the findings and address potential selection biases.
- Analyze and discuss the unusually long extubation times observed in both groups, as this may relate to delayed bowel recovery.
- Where possible, compare “days to drain removal” instead of total drainage volume, as this may provide a more meaningful measure of recovery.
- Discuss intraoperative vascular injuries and clarify the proportion of extended right hemicolectomies in each group, particularly in relation to lymph node count differences.
- Highlight the apical lymph node metastasis rate (14.9%) for tumors above the ileocecal valve (ICV) and discuss its implications for D3 dissection.
- Provide additional details for the two cases with lymph node metastasis in the anterior tissue of the SMV, including: Clinical TNM staging (cTNM), Tumor location, Evidence of skip metastases, and Preoperative CT findings.

The manuscript currently focuses on immediate pathological and surgical outcomes. To improve generalizability and impact, consider including long-term survival outcomes and their correlation with the techniques compared.

We look forward to receiving your revised submission.

·

Basic reporting

The article is well-written, and the content is explicit. However, Figure 1 is not referred to in the main text. Also, in Figure 1, sections E & F have not been labeled.

The authors also focus on the tissue anterior to the superior mesenteric vein (SMV) and the presence of neural tissue in it. The raw data provided, however, doesn’t mention this aspect. This is the most relevant data in the paper, and it is missing.

Further, the raw data mentions that CA 19-9 values are dichotomous (below and above 35 ng/ml). There is no explanation of how this value was reached.

Experimental design

The study is a retrospective analysis and thus fails to provide valuable evidence concerning the scientific question raised by the authors in a manner that can be generalized.

I would, however, encourage the authors to perform propensity-matched analysis for the data and match the data for age, gender, stage of the primary tumor, and location of the primary tumor, and then determine if there is any association between the two methods of lymphadenectomy.

Further, in the analysis, no correlation is provided to determine if the presence/absence of neural tissue in the tissue anterior to SMV altered the short-term outcomes. This would be the most relevant point for this article since the authors hypothesize that the disruption of the neural structure in this region results in delayed bowel recovery.

Validity of the findings

The authors have mentioned that they had a change in the method of lymph node dissection post-2019, and they did this because a meta-analysis of randomized trials on central vascular ligation had shown survival benefits. Their paper only focuses on immediate pathological and surgical outcomes and doesn’t provide long-term survival outcomes. Thus, generalizing their findings would be difficult without this crucial data due to the glaring survival outcomes from CME surgical trials.

I strongly urge the authors to relook at these long-term outcomes and correlate them.

·

Basic reporting

I thank the authors for their work. This is a retrospective case-control study comparing two lymph node dissection techniques (D2 vs D3) after oncological right colectomies.
- English is acceptable
- References are up-to-date and the authors did cite the most important studies re this subject.

Experimental design

A few comments:
- The postop extubation time is 3.3 vs 5.3 days (p<0.001). This is unusually long for both groups. Maybe the prolonged intubation time (longer in the SMV left group) could explain the slower bowel recovery? Maybe propensity score matching using extubation time as a dependent variable could clarify the potential risk of selection bias?
- There is a significantly higher volume of drainage in the SMV-left group. This should be discussed briefly. Also, the total volume doesn`t really mean much from a recovery perspective, but rather days to drain removal does. Can the authors compare days-to-drain removal? This is optional, as hospital stay was measured and was similar in both groups, which is the most important.
- Did the authors compare intraoperative vascular injuries?
- How many were extended right hemicolectomies between the two groups? The higher lymph node count could be explained by a higher proportion of MCA group dissection, which the authors should clarify.

Validity of the findings

- The authors emphasize the higher risk of postoperative complications in the SMV-left group, but this is marginally true, as individual complications (especially postop bleeding and chile leak, which one would be worried about) are similar. So I would be more reserved when calling out the SMV-left group as having more complications.
- To me, an important highlight of the article is the 14.9% apical lymph node metastasis rate for tumours located above the ICV. For this reason only, I would encourage D3 in all cases with lesions above the ICV. What is the apical lymph node metastasis in tumours above the ICV if we include inly SMVl-eft patients?
- "Notably, in the SMV-left group, two cases of lymph node metastasis were found in
235 the anterior tissue of the surgical trunk of the superior mesenteric vein (SMV)". What was the cTNM of these patients? What was the location of the tumour? Were there skip metastasis in these two cases? Did the preoperative CT depict any + nodes around the SMV?

Additional comments

no comment

---

## Round 0.2 · Major Revisions

Dear authors,

Thank you for your submission. However, I believe that important aspects still require revision. Please, consider adding to the manuscript key explanations such as for using 35 ng/ml as the dichotomous cutoff for CA 19-9 values, and explanation about extubation times (other readers may have the same doubts, so where possible and relevant explanations should also have been added to the manuscript) ... and refer to the reviewers' comments for further detail.

·

Basic reporting

Unambiguous, professional English is used except in the revised portions.

Raw data still has the key component missing - how many nodes from the anterior tissue of the SMV surgical trunk were positive for metastases on histopathology? The entire article is based on the premise that dissection of the tissue anterior to the SMV surgical trunk is unnecessary since lymph node metastases are rare. However, corroborative raw data has been provided to this effect.

Even in Figure 2, where data is depicted regarding the total and positive nodes at each level, data regarding the nodes in the tissue anterior to the SMV trunk is missing.

Experimental design

I commend the authors for having performed the propensity score-matched analysis. It would have been more relevant to match the other possible causes of prolonged post-operative course, i.e., primary tumor stage.

Validity of the findings

The authors have not analyzed the other reasons for delayed bowel recovery by performing logistic regression. This is one of their strong reasons for recommending no dissection anterior to the SMV. The mere presence of nerve fibers doesn't justify the delayed bowel recovery.

Regarding skip metastases, on preliminary analysis of the raw data provided, I could decipher that four patients in the SMV right group ( patient ID 27,31,64 and 85) and three patients in the SMV - left group had skip metastases to the main lymph node (104,113,120). This needs to be looked into.

Additional comments

I would advise the authors to provide detailed information regarding the objectives being addressed.

·

Basic reporting

Suitable for publication now

Experimental design

Suitable for publication now

Validity of the findings

Suitable for publication now

Additional comments

Suitable for publication now

---

## Round 0.3 · accepted · Accept

I am now approving your manuscript for publication. Thank you! Congratulations.

·

Basic reporting

The reporting is clear and unambiguous.

Experimental design

The authors have made the necessary amendments to the study design.

Validity of the findings

The findings have been further validated.

Additional comments

I thank the authors for considering the recommendations and also for making the necessary amendments.